# The Role of miRNAs to Detect Progression, Stratify, and Predict Relevant Clinical Outcomes in Bladder Cancer

**DOI:** 10.3390/ijms25042178

**Published:** 2024-02-11

**Authors:** Maria Iyali Torres-Bustamante, Jorge Raul Vazquez-Urrutia, Fabiola Solorzano-Ibarra, Pablo Cesar Ortiz-Lazareno

**Affiliations:** 1Centro Universitario de Ciencias de la Salud (CUCS), Universidad de Guadalajara, Guadalajara 44340, Mexico; iyali@hotmail.es; 2Department of Medicine, The Pennsylvania State University College of Medicine, The Pennsylvania State University, Hershey, PA 17033, USA; jvazquezurrutia@pennstatehealth.psu.edu; 3Instituto de Investigación en Enfermedades Crónico Degenerativas, Departamento de Biología Molecular y Genómica, Centro Universitario de Ciencias de la Salud (CUCS), Universidad de Guadalajara, Guadalajara 44340, Mexico; fabiolasolorzanoibarra@gmail.com; 4Estancias Posdoctorales por México, Consejo Nacional de Humanidades, Ciencias y Tecnologías (CONACYT), México City 03940, Mexico; 5División de Inmunología, Centro de Investigación Biomédica de Occidente, Instituto Mexicano del Seguro Social, Guadalajara 44340, Mexico

**Keywords:** miRNAs, bladder cancer, biomarkers, prognosis

## Abstract

Bladder cancer (BC) is one of the most common types of cancer worldwide, with significant differences in survival depending on the degree of muscle and surrounding tissue invasion. For this reason, the timely detection and monitoring of the disease are important. Surveillance cystoscopy is an invasive, costly, and uncomfortable procedure to monitor BC, raising the need for new, less invasive alternatives. In this scenario, microRNAs (miRNAs) represent attractive prognostic tools given their role as gene regulators in different biological processes, tissue expression, and their ease of evaluation in liquid samples. In cancer, miRNA expression is dynamically modified depending on the tumor type and cancer staging, making them potential biomarkers. This review describes the most recent studies in the last five years exploring the utility of miRNA-based strategies to monitor progression, stratify, and predict relevant clinical outcomes of bladder cancer. Several studies have shown that multimarker miRNA models can better predict overall survival, recurrence, and progression in BC patients than traditional strategies, especially when combining miRNA expression with clinicopathological variables. Future studies should focus on validating their use in different cohorts and liquid samples.

## 1. Bladder Cancer Overview

Bladder cancer (BC) represents the tenth most common cancer in the world and the fourth most common cancer in men in the US [1,2]. Several risk factors associated with the development of this disease have been described, the most well established being tobacco smoking, which is correlated with almost 50% of newly diagnosed BC cases. Other risk factors include the occupational exposure to aluminum, rubber, painting, dyes, arsenic, environmental radiation, or parasitic schistosomal infections [2]. Recent studies have found that the urobiome, the presence of biofilm-associated bacteria (Porphyromonas) responsible for chronic inflammation, can promote carcinogenesis [3].

BC’s most common clinical presentation is gross or microscopic hematuria and irritative voiding symptoms, such as dysuria, urgency, and frequency. On other occasions, the tumor is accidentally discovered on imaging [4]. The American Urologic Association recommends cystoscopy for bladder evaluation and endoscopic resection for patients with a gross hematuria in the appropriate clinical context. For those with microscopic hematuria, the decision is mainly based on the presence of high-risk factors [5]. In addition, a CT urogram usually follows diagnosis to assess the axial extension of the disease.

BC generally originates from the bladder urothelium and is referred to as urothelial carcinoma; almost 90% of these are transitional cell carcinomas. Based on the degree of invasion, BC is divided into non-muscle-invasive bladder cancer (NMIBC) and non-papillary muscle-invasive bladder cancer (MIBC) [6].

Patients with BC are classified by their tumor grade and by their tumor stage using the eighth edition of the Staging Manual by the American Joint Committee on Cancer (AJCC) [7] Appendix A. NMIBC generally encompasses Tis tumors to the T1 stage and low-grade tumors, while T2–T4 and high-grade tumors are included within the MIBC group. NMIBC represents 75–85% of all new BC cases diagnosed.

It is important to note that, aside from the morphological distinction between NMIBC and MIBC, these types of BC have different prognoses, treatments, and clinical outcomes. NMIBC is generally managed with the transrectal resection of the tumor (TURPBT) and has a 5-year probability of recurrence and a progression of 78% and 45%, respectively [8]. After diagnosis and initial treatment, a 3–4-month surveillance cystoscopy is recommended. Then, based on the clinicopathologic data such as the tumor grade, stage, and response to initial treatment, patients with NMIBC are risk-stratified, which determines the cystoscopy frequency following the first post-treatment negative cystoscopy [5].

MIBC accounts for the rest of cases. These are often managed with radical cystectomy with chemotherapy. However, despite treatment, the 5-year cancer-specific survival for MBIC is 60%, and about 50% develop metastatic disease [9].

Despite its efficiency in the early detection of tumor progression or recurrence, surveillance cystoscopy is an invasive, costly, and uncomfortable procedure. Furthermore, the frequency of surveillance cystoscopies makes NMIBC the most expensive cancer to monitor [10]. In low-risk patients, data show that the overuse of these strategies greatly increases healthcare costs [10,11]. For these reasons, developing new, less invasive, and convenient methods for surveillance is needed.

In this sense, the concept of a liquid biopsy involves the use of biological fluids to assess the cancer-derived components. Liquid biopsy has inarguable potential as an attractive alternative to evaluate tumor profiling. Liquid biopsies are less invasive, more accessible, highly sensitive, and can provide real-time information on cancer progression and treatment response. Using liquid biopsies allows for the evaluation of whole tumors without being limited by their heterogeneity, which increases the amount of biopsies needed to assess the totality of the tumor. Multiple components can be measured in liquid biopsies, including the circulating tumor cells, DNA or RNA-based molecules, and extracellular vesicles, which have been one of the main focuses of cancer research [12,13].

The only recommended biomarker in the guidelines to monitor BC is urine cytology, which can be used along with cystoscopy. However, it has a low sensitivity (average, 48%) that can be even lower in low-grade tumors [14]. More precise biomarkers could improve the detection of progression and recurrence, reduce the need for cystoscopy, and improve risk classification systems’ performance. Other FDA-approved alternatives to urinary cytology for the initial detection and surveillance of BC include the nuclear matrix protein 22 (NMP22) kit, NMP22, and UroVysion, which are also mentioned in the European Urological Guidelines, the BladderChek Test, BTA-TRAK and BTA stat kits, and Cell Search. However, despite having similar or superior performance compared to cytology, these have varying degrees of sensitivity and specificity, are limited to specific patient scenarios, and are limited in their applications in clinical practice [15].

A biomarker requires certain qualities before it may be used to indicate disease. Firstly, it should be easily obtained and cost-effective, avoiding complicated processing methods that could hinder generalizability. Secondly, the biomarker must exhibit a sensitivity and specificity for the targeted condition. Lastly, it should be non-invasive to minimally invasive to prioritize patient comfort and reduce potential complications [16]. In this context, microRNAs (miRNA) become potential biomarkers in cancer.

MicroRNAs (miRNA) are 19–25 nucleotide non-coding RNAs generated from DNA templates through RNA polymerase III. Shortly after transcription, they are modified by the enzyme Drosha to be exported to the cytoplasm to be cleaved by the enzymes Dicer or Ago2 to form the effector complex RNA-inducing silencing complex (RISC). Through this complex, they bind to the 3 ‘untranslated regions’ of their target mRNA through base pairing, where they can exert their consequent functions based on the nature of complementarity, where a near-perfect pairing results in degradation and a moderate pairing results in translation inhibition without cleavage. Overall, this results in translation inhibition, silencing, and gene expression inhibition [17,18].

Recently, it has been shown that some miRNAs can enhance the gene expression of their respective targets [18]. It is worth noting that one miRNA typically targets multiple mRNAs, whereas multiple miRNAs can regulate a single mRNA, making them indirect indicators of the expression of many genes and proteins that are usually involved in similar biological processes [19].

The processes controlled by miRNAs in normal cells include proliferation, cell development, and apoptosis. They also regulate hematopoiesis, bone formation, gastrulation, muscle, and neural development. In cancer and other diseases, miRNA biogenesis is altered, impacting their expression, and making them surrogate biomarkers for carcinogenesis. On the other hand, they have a direct role in the development of cancer and its progression, affecting the expression of both oncogenes and tumor suppressor genes in a tumor-specific fashion. For example, miR-125b is upregulated in ovarian, thyroid, breast, and oral squamous-cell carcinomas, where it has a tumor suppressor role in inhibiting cell proliferation and cell-cycle progression [20,21,22]. At the same time, miR-125b in prostate, thyroid, glioblastoma, and neuroblastoma cancers acts as a protooncogene by inhibiting apoptosis-promoting cell proliferation and invasion in a p53-dependent manner [23,24].

MiRNAs are, per se, very stable compared to other RNA-based molecules, given their small size and binding properties [25]. miRNAs can be secreted in the tumor microenvironment, and bound with circulating proteins or exosomes, which makes them easily accessible in bodily fluids. For these reasons, miRNAs are promising candidates for cancer detection and monitoring.

In the oncology field, microRNAs have been extensively studied in different malignancies, where they have proven to be reliable markers for early diagnosis, stratification, and even treatment options. For example, in pancreatic cancer, a cancer that is often late diagnosed, several studies have proven that miRNA panels have superior diagnostic performances compared to CA-19-9, one of the two only FDA-approved biomarkers for this malignancy [26,27]. Similar findings have been discovered in diseases like leukemia, colon cancer, and breast cancer [26]. In the case of colorectal cancer, miR-517a is a tumor-associated miRNA that has a role in cell migration and invasion; it also can be used as a prognostic marker for predicting survival [28]. In ovarian cancer, miR-532-5p works as a tumor suppressor and a high expression is associated with a better prognosis [29]. In bladder cancer, different studies support the potential use of miRNAs as diagnosis, prognostic, and response to treatment indicators [30,31,32,33,34]. For these reasons, this review describes the most recent studies exploring the utility of miRNA-based strategies to detect progression, stratify, and predict relevant clinical outcomes in bladder cancer.

## 2. miRNAs as Prognostic Tools in BC

An initial search in the PubMed database using the strings ″miRNAs” OR “microRNAs” AND “bladder cancer” AND “prognostic” as well as “miRNAs” AND “bladder cancer” AND “treatment response” yielded 371 articles. The included studies were those utilizing human samples, evaluating prognostic outcomes such as progression, survival, or recurrence, and published within the last five years, as this is a rapidly advancing field that requires up-to-date results. Other reviews and metanalysis were excluded as well as those using cellular or animal data only. Studies focusing on diagnosis or screening were also excluded. After applying these criteria, 37 original studies were included in this literature review. The study selection using the PRISMA flowchart is shown in Figure 1. Of note, all these studies are retrospective and used real clinical data for their analysis [35,36,37,38,39,40,41,42,43,44,45,46,47,48,49,50,51,52,53,54,55,56,57,58,59,60,61,62,63,64,65,66,67,68,69,70,71].

We further subdivided these studies according to their use of miRNAs as markers of relevant clinical outcomes such as recurrence or survival [Table 1], progression from NMIBC to MIBC or a low to high tumor stage [Table 2], as well as their ability to stratify patients among groups according to specific miRNA-based models [Table 3].

Different outcomes were evaluated among the studies, the most common being the survival and recurrence; however, some studies analyzed other specific prognostic endpoints. The way this was assessed was, as mentioned, based on the level of differential expression among groups. For example, Setti et al. demonstrated that increased miR-9 tissue expression was higher in patients with MIBC compared to NMIBC patients. Furthermore, its expression was higher in high-grade NMIBC than low-grade NMIBC patients, demonstrating being a potential biomarker to distinguish between them [57]. Similarly, Awadala et al. found that decreased levels of tissue let-7a-5p, miR-449a-5p, -124-3P, and -138-5p, and increased miR-23a-5p were correlated with muscle invasion, respectively [36].

Another relevant prognostic outcome is treatment response, which directly influences patient prognosis. As such, Khan et al. used a 14-miRNA tissue-based signature that classified patients with MBIC in a hypoxic and non-hypoxic phenotype, an independent prognostic for treatment selection, that was demonstrated to be a responder modifier to hypoxia-modifying therapy in several cohorts. Greater accuracy was achieved when combining this score with an mRNA score developed by the same group [45]. In another study, it was shown that in a sample of NMIBC patients with no response to Bacillus Calmette-Guérin (BCG) therapy, there was an increased expression of miR-199a and miR-21 as well as a decreased expression of let-7a and miR-31 [37].

As previously mentioned, an ideal biomarker should come from a non-invasive source, like blood, urine, or serum. Of the analyzed studies, only six used liquid biopsy measurements to compare between patient groups [35,41,49,58,67,69]. In one study, Andrew et al. demonstrated that in patients with NMIBC, high miR-26b-5p tissue expression was associated with a longer time to recurrence and that combining this molecule with classic clinical variables such as the tumor size, stage, and grade increased the area under the curve (AUC) to predict the tumor recurrence [35]. It should be noted that this study did not demonstrate a correlation of tissue miRNA expression with urine and blood. In contrast, Cavallari et al. considered urine an ideal liquid biopsy to assess BC for all aspects; however, since hematuria is one of the most common presenting symptoms, miRNAs measured from these samples may derive from RBCs and not from cancer cells. As such, they isolated non-RBC-derived miRNAs. With this, they developed a decision tree composed of miR-34a, miR-200a, and miR-193a that was able to accurately classify high- and low-risk patients for progression, identified by their European Urologic Association risk score, with a sensitivity of 84% and specificity of 82% [41].

Additionally, while the expression of these miRNAs differed between groups in urine, it did not in blood, suggesting that miRNAs released by cancer cells can be measured in urine but not in blood. Lin et al. developed a serum 7 miRNA prognostic signature (miRNA-185-5p, -66a, -30c-5p, -3648, -1270, -200c-3p, and -29c-5p) that classified BC patients as high vs. low risk, with the high-risk group having worse overall survival [49]. Furthermore, Yang et al. and Zaidi et al. demonstrated that an increased expression of miR-10 in urine and serum was correlated with tissue expression and accurately differentiated between NMIBC and MIBC. miR-10 was associated with an increased tumor grade and stage [67,69]. Similarly, in the study by Shee et al., let-7f-5p expression in tissue, plasma, and urine was correlated, and increased levels were associated with longer recurrence-free survival [58].

It is also noteworthy that most studies classified BC patients based on the expression level of the analyzed miRNAs, which is useful when addressing their implications. The most reported miRNAs in the different studies were miR-141 and -200, which were both members of the miR200 family in seven studies [39,41,51,62,63,65,68]; and according to the results, a high expression of these molecules was correlated with higher overall survival [39,51,62,65,68] and lower MIBC incidence [63]. Interestingly, these studies were performed on tumor tissue samples, while the only study using liquid biopsies showed that increased urine miR-200a levels were correlated with high-risk BC [41]; this could be explained by methodological variability amongst the studies. Some other frequent miRNAs reported include the Let-7 family in four studies [36,58,66,68], miR-100-5p in two studies [38,43], miR-138-5p in two studies [36,38], and miR-205 in two studies [39,40], respectively. This is relevant because consistency between studies in selecting the most appropriate signatures should be considered to incorporate biomarkers in clinical practice.

It is a well-known fact that multimarker-based models outperform single-molecule measurements because, as previously noted, the results may conflict with other studies, which limits their applicability and tends to remain below classic clinicopathological variables for prognostic purposes. On the contrary, Liu et al. developed a tissue gene, long non-coding mRNA, and miRNA score that had higher AUC for survival compared to the TNM stage [52]. Likewise, the tissue 7 miRNA score of Xv et al. better predicted overall survival in patients with BC compared to clinicopathological variables [66].

Some other studies demonstrated the value of adding miRNAs to clinical models to increase their performance. As such, the Andrew et al. study showed that to estimate recurrence, including the results of tissue miR-26-5p levels together with clinical factors (such as sex, age, stage of TNM, and tumor grade), increased the AUC for recurrence compared to either alone [35]. Similarly, Juracek et al. studied and validated in different cohorts that adding miR-34a-3p tissue levels to the EORTC nomogram (a tool to predict recurrence or progression of NMIBC) increased the SE and SP to predict tumor recurrence in NMIBC patients [44]. In another study, Urabe et al. constructed and validated a model combining lymph node invasion and tissue miR-23a-3p, -3679-3p, and -3195 expression that had an AUC of 88%, SE of 87%, and SP of 30% for tumor recurrence and was associated with overall survival [61]. Furthermore, Xiong et al. also constructed a clinical-mRNA–miRNA tissue signature with a higher AUC for overall survival than alone [65].

## 3. Biological Plausibility of Described miRNAs in BC

To further support the incorporation of miRNAs in bladder cancer, there must be a concordant mechanism that explains the changes in their expression during different tumor stages. We focused on the most consistently reported miRNAs across the analyzed studies [Figure 2].

As described in two of the analyzed studies, miR-100 has been described as an antitumoral miRNA in most cancer-related studies, and it demonstrated a similar role in BC. [72]. As such, Blanca et al. showed that increased miR-100 expression was correlated with a decreased expression of the protooncogene *FGFR3*. Interestingly, the study also associated decreased levels of tissue miR-100 with better clinical outcomes [38]. Similarly, the miRNA score used by Inamoto et al. showed a decreased expression of tissue miR-100 in NMIBC compared to MIBC patients [43]. At the same time, Blanca et al. also found a marginally significant relation between miR-100 downregulation and NMIBC. However, previous studies have found an inverse clinical correlation with miR-100 downregulation predicting poorer clinical outcomes, which is more in line with its putative antitumoral role [72].

Another frequently reported miRNA was miR-138-5p, also described as a tumor suppressor miRNA in BC. In line with this, the study of Awadala et al. found decreased levels of this miRNA in pT2-pT4 compared to pT1, which correlated with decreased cancer-specific survival [36]. They also found a negative correlation between tissue miRNA-138-5p expression and the levels of *HIF-1a*, a protooncogene involved in cancer progression at promoting vasculogenesis [36]. On the other hand, Blanca et al. observed that a higher tissue expression of miRNA-138-5p in NMIBC patients correlated with better recurrence-free survival, and it was associated with increased levels of cyclin D3 protein expression, which is contrary to the tumor suppressor role of this miRNA [38].

Studies by Borkowska et al. and Braicu et al. found that the overexpression of miR-205 in BC was correlated with worse clinical outcomes, particularly in pT2 relative to T1aTa tumors [39,40]. Braicu et al., using Ingenuity Pathway Analysis from data obtained on next-generation sequencing from tumor tissue samples, showed that miR-205 indirectly influenced *AKT,* an important protooncogene. In addition, miR-205 inhibited important tumor suppressor genes, including *Rb*, *P53*, and *E-cadherin*, but also some protooncogenes like *VEGF*, *MMPs*, and *Ras*, which displays a heterogeneous nature of miRNA-target interaction [39].

The Let-7 miRNA family (a, b, c, d, e, f, g, i, and miR-98 and miR-202) was mentioned and validated across several studies [36,58,66,68]. In the study by Shee et al., high levels of Let -7f-5p in tumor tissue and blood in multiple cohorts were associated with less tumor recurrence. BC cells expressing high levels of let-7-5p showed decreased viability and migration and a reduced expression of the target *HMGA2* gene, which is implicated in cell migration [58]. Likewise, Awadalla et al. observed that let-7a-5p expression levels were decreased in higher tumor-stage tissues and were inversely correlated with the expression of *FZD4*, *WNT7A*, and *b-Catenin* genes involved in the Wnt b-Catenin pathway, which is relevant for cancer progression. Furthermore, low levels of let-7a-5p miRNA and high levels of *FZD4*, *WNT7A*, and *b-Catenin* genes were correlated with worse cancer-specific survival [36].

Lastly, the members of the miR200 family (miR-200a, -200b, -200c, -141, and -429) were the most analyzed in the reviewed studies [39,41,51,62,63,65,68]. These miRNAs have been described as tumor suppressors in some cancers. In line with this, Braicu et al. described an inverse relation between the levels of tissue miR-200c and miR-141 and the expression of *ZEB1*. This important protooncogene downregulates *E-Cadherin* expression, a crucial component of the epithelial mesenchymal transition [39]. Moreover, Liu et al., through an mRNA–miRNA–lncRNA interaction bioinformatics approach, showed that *GATA3*, a tumor suppressor gene, was positively regulated by miR-141 [51]. Ware et al., through a gene expression analysis, showed that miR-200c and 141 targeted genes are involved in cancer progression; for example, *KDR* is involved in vasculogenesis, and *ZEB1* promotes the epithelial–mesenchymal transition. There has been shown to be a gradual decrease in miR-200c and 141 expressions from the T1 to T4 tumor stages [62]. All these findings were supported in the remaining studies [63,65,68], establishing the clear role of the miR-200 family as tumor suppressors in BC.

The most common risk factor for BC is smoking, and some studies have implied that tobacco smoking can alter miRNA expression. Ware et al.’s study showed significant differences in the levels of miR-200c and miR-141 expression in different degrees of smoking in patients with BC [62]. This is relevant as the correlation of these molecules with cancer risk factors is essential to creating a rationale for their use as markers of disease.

## 4. Limitations of miRNA-Based Strategies for BC

Despite the encouraging results, some limitations may hinder the application of miRNA-based prognostic strategies for BC. The first limitation was the type of biopsy used, tissue rather than liquid, in several studies, causing a wedge between the studies and the clinical practice, consequently decreasing the cost-benefit justification, and limiting its applicability. Moreover, it has been demonstrated that BC is a disease with considerable tissue and genetic heterogeneity, which can affect miRNA profiles; it could account for some of the contradictions noted in tissue-based studies [12].

It is also important to mention that miRNAs can be identified from different matrices due to their stability and resistance to storage handling. For example, miRNAs can be preserved in serum for 10 years [73], which justifies the evaluation of these molecules in serum and other biological fluids, including urine. However, the origin of miRNAs depends on the biological source; in the case of miRNAs evaluated in serum, they can be derived from circulating blood cells and cancer cells. This should be considered when selecting the best miRNA signature with clinical applications that differentiate diseases from healthy states. In addition, evaluating exosomes containing miRNAs is an interesting way to analyze these molecules, as tumor-secreted exosomes modulate other cells from a distance, representing a rationale underlying their evaluation in fluids as surrogates from the tumor microenvironment [74].

It is worth noticing that most studies had a limited sample, which decreased their power to detect significant results. Regarding the methodological aspects, techniques used to evaluate miRNA analysis should be considered to compare the results from different studies. The most commonly used are microarrays, quantitative real-time PCRs, and next-generation sequencing [21]. Various techniques were used for the analysis of BC, as some studies used microarrays, others PCRs, and others bioinformatics-based approaches. Interestingly, Wei et al. proposed the use of PhC barcodes with a hybridization chain reaction (HCR), a reliable method that, in contrast with microarrays, can change the positions in the detection solution randomly without affecting the coding effect and provides a larger surface for probe ligation, allowing for multiple-biomarker detection. Moreover, HRC is enzyme-free and uses isothermal amplification, eliminating the temperature variation required for PCR [64]. Standardizing protocols for sample collecting and processing are also needed to avoid limitations in comparing different studies to select miRNA profiles with clinical applications.

In addition, the data normalization method and reference molecules varied amongst studies, which could partly explain the variability in the expression and discovery of certain miRNAs, as some studies report upregulation and downregulation of the same molecule, and most of the studies reported miRNAs unrelated to one another. Moreover, some miRNA expression profiles did not correlate with gene expression, which is relevant to support their clinical usage. Selecting adequate reference genes to normalize the miRNA levels is important to obtain comparable results between studies [75].

Finally, studies must have consistent results, which can be evaluated by their clinical validity, which is an important characteristic in cancer biomarker research that refers to the ability of a biomarker to classify a sample into two groups, which is further reinforced when the biomarker is applied to independent patient cohorts [12]. In this review, few of the analyzed studies validated their findings [35,39,42,43,44,45,46,53,58,60,67], and only one of them used liquid biopsy as a source to evaluate miRNAs [67].

## 5. Future Perspectives

From the analyzed studies, the most promising miRNAs were the members of the miR-200 and Let-7 families, not only for being frequently mentioned and included in prognostic models with good performance [39,41,65,66] but also for their biological plausibility, as their increased expression correlated with better clinical outcomes, which was consistent with their influence in cancer-related pathways. Between these miRNAs, let-7f-5p expression in tissue and liquid biopsy was correlated with appropriate clinical outcomes and had a plausible mechanism supporting the observed deregulation, making this molecule one of the best-analyzed miRNAs for being truly non-invasive [58]. Therefore, studying these miRNAs could yield more value to miRNA-related research in BC.

It is worth noticing that although all the validated multimarker scores were developed using tissue miRNA expression [43,53,60], future studies could strengthen the use of these scores in liquid biopsy samples and different cohorts [Figure 3].

Different techniques, including microarrays, quantitative real-time PCRs, and next-generation sequencing, are used to evaluate miRNAs. However, a standardized protocol would help to avoid limitations during the comparisons between different studies. Thus, improving the selection of miRNA profiles can allow for diagnosing or predicting responses to treatment or survival. A greater number of studies of these molecules in urine samples may help to find a specific signature for bladder cancer patients.

## 6. Conclusions

miRNAs are important molecules in regulating gene expression. In cancer, an alteration in their expression is observed, and they can act as tumor suppressors or promoting molecules, depending on the tumor type. Due to their stability and changes in expression during a disease, they are promising biomarkers. This study evaluated the use of miRNAs as prognostic tools in bladder cancer. Despite the limitations needed to project these to clinical use, some studies have shown their potential to surpass clinicopathological variables in predicting the overall survival, recurrence, and progression from NMIBC to MIBC when used as multimarker models, and to increase the efficiency of these data when added together.

Therefore, important steps are needed to validate their use further: first, increasing sample sizes and validating results in multiple cohorts; second, using multimarker models, with the specific aim of clinical-miRNA models that could be easier to translate in clinical practice; and finally, a consensus should be reached regarding the technique for analysis and source extraction as, although easily obtained, tissue-derived miRNA studies do not address the issue of invasiveness. Future studies should focus on consistent, reproducible, and non-invasive miRNAs to develop and test the already-established models to improve their performance for prognostic purposes.

## Figures and Tables

**Figure 1 ijms-25-02178-f001:**
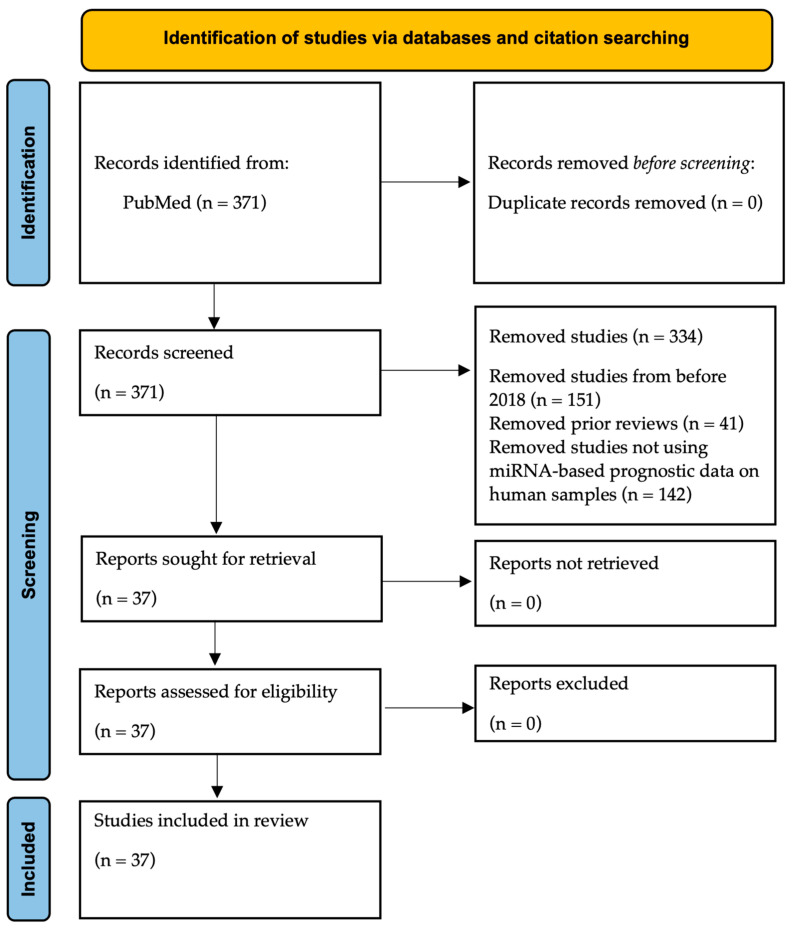
PRISMA flow diagram of the included studies in this review.

**Figure 2 ijms-25-02178-f002:**
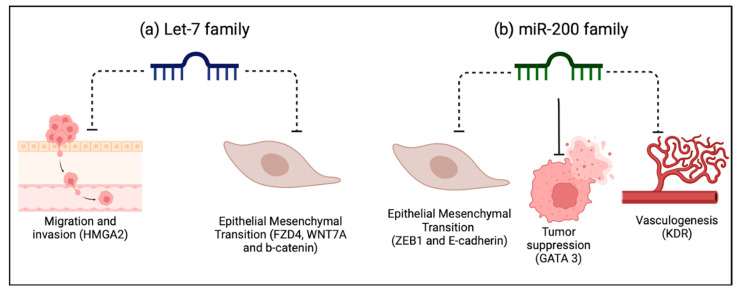
Biological plausibility of the most promising miRNAs in BC. Considerations are based on the analyzed studies. (**a**) The Let-7 family is involved in the negative regulation of the process of cell migration and invasion, as well as in epithelial mesenchymal transition, fundamental processes in cancer. (**b**) The miR-200 family regulates GATA3, acting as a tumor suppressor by directly inhibiting epithelial-to-mesenchymal transition (EMT). It also negatively regulates vasculogenesis, and both these processes are fundamental for metastasis. Dotted lines represent inhibition. Solid lines represent an induction of expression.

**Figure 3 ijms-25-02178-f003:**
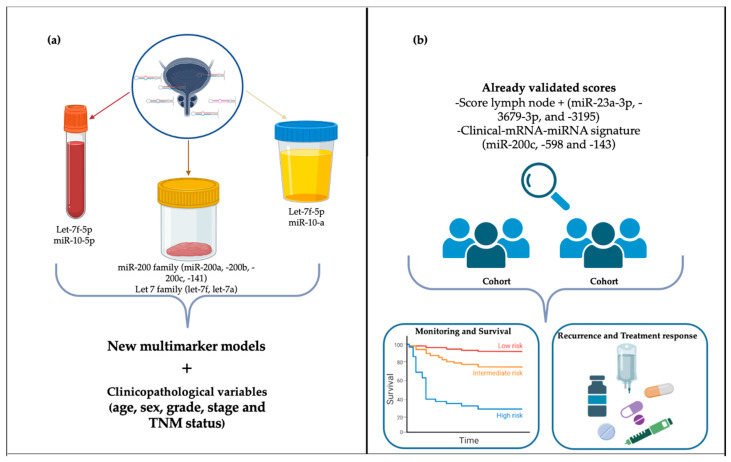
Future perspectives for miRNAs as prognostic markers in bladder cancer. (**a**) Construction of future models based on validated and biologically plausible miRNAs combined with clinicopathological variables. (**b**) Further testing already validated models in larger cohorts to increase the prognostic potential and clinical applications.

**Table 1 ijms-25-02178-t001:** Studies identifying miRNAs as markers for clinical outcomes of BC.

miRNAs Analyzed	Performance	Group Comparison	Type of miRNA	Source	Technique for Analysis	Sample Size	Validation	Authors and Year
miR-26b-5p	↑ miR-26-5p = ↑ RFS miR-26-5p + BF = ↑ AUC of BF alone for recurrence	Low expression vs. High expression	Free and exosomal	Tissue, blood and urine	Microarrays	231	Yes	Andrew 2019 [35]
miR-21, -199, -31, let-7a	↑ miR-21, -199 and ↓ miR-31, let-7 in BGC non responders ↑ miR-21, -199 and ↓ miR-31, let-7 = ↓ RFS	NMIBC BCG responders vs. non-responders	Free	Tissue	RT-QPCR	157	No	Awadalla 2022 [36]
miR-138-5p and miR-100-5p	↑ miR-138-5p in LGT ↓ miR-138-5p in recurrent tumors ↑ miR-138-5p = ↑ RFS ↓ miR-100-5p = ↑ RFS and ↑ CSS	Low expression vs. High expression	Free	Tissue	RT-QPCR	50	No	Blanca 2019 [38]
miR-205-5p, -20a-5p, -21-5p, -145-5p and -182-5p	↑ miR-205-5p, -145-5p, and -21-5p = ↑ risk of death ↑ miR-20a-5p and -182-5p = ↑ risk of recurrence	Stage	Free	Tissue	RT-QPCR	85	No	Borkowska 2019 [40]
miR-143, -139, -141, -205 and -23a	↑ miR-141 and ↓ miR-143 = ↑ OS	Low grade vs. High grade	Free	Tissue	Microarrays, RT-QPCR and TCGA analysis	450	Yes	Braicu 2019 [39]
miR-30c-5p	↓ miR-30c-5p = Poor prognosis	Low expression vs. High expression	Free	Tissue	RT-QPCR and TCGA analysis	445	Yes	Hao 2023 [42]
miR-34a-3p	↓ miR-34a-3p = ↑ OS miR-34a-3p + EORTC nomogram = ↑ SE and SP for progression	Low expression vs. High expression	Free	Tissue	Microarrays and RT-QPCR	137	Yes	Juracek 2019 [44]
miR-106b-5p	↑ miR-106b-5p = ↑ OS	Low expression vs. High expression	Free	Tissue	TCGA and Choi analysis	1071	Yes	Lee 2018 [46]
miR-302-b	↓ miR-302-b = ↓ RFS	Low expression vs. High expression	Free	Tissue	RT-QPCR	39	No	Li 2018 [47]
miR-187-5p	↑ miR-187-5p = ↑ Recurrence risk	Low expression vs. High expression	Free	Tissue	RT-QPCR	44	No	Li 2018 [48]
miR-325	↑ miR-325 = ↓ OS	Low expression vs. High expression	Free	Tissue	RT-QPCR	164	No	Lin 2018 [50]
miR-141-5p, -141-3p and -200c-3p	↑ miR-141-5p, -141-3p and -200c-3p = ↑ OS	Low expression vs. High expression	Free	Tissue	TCGA analysis	403	No	Liu 2018 [51]
AGO1, AGO2 and Drosha	↑ Drosha = ↑ OS	Low expression vs. High expression	Free	Tissue	Microarrays	112	No	Rabien 2018 [56]
Let-7f-5p	↑ Let-7f-5p = ↑ RFS	Low expression vs. High expression	Free and exosomal	Tissue, blood and urine	NanoString’s amplification	207	Yes	Shee 2020 [58]
miR-211-5p	↓ miR-211-5p = ↓ OS and ↑ TNM stage	Low expression vs. High expression	Free	Tissue	Microarrays and RT-QPCR	58	No	Wang 2020 [61]
3 Clusters (miR-200c/miR-141) (miR-216a/miR-217) (miR-15b/miR-16-2)	↑ (miR-200c/miR-141) = ↑ OS ↑ (miR-216a/miR-217) = ↓ OS	Degree of expression among BC patients	Free	Tissue	Cluster miRNA analysis TCGA analysis	412	No	Ware 2022 [62]
miR-429	↓ miR-429 = ↓ 5-year OS and RFS	Low expression vs. High expression	Free	Tissue	In situ hybridization	76	No	Wu 2018 [64]
miR-432	↑ miR-432 = ↑ OS and ↑ DFS	Low expression vs. High expression	Free	Tissue	RT-QPCR	156	No	Zhang 2021 [70]
miR-195	↑ miR-195 = ↓ OS	Low expression vs. High expression	Free	Tissue	TCGA analysis	418	No	Zhu 2018 [71]

Note: Only statistically significant results are shown. Highlighted studies reflect those using multimarker models. BC = Bladder Cancer, RFS = Recurrence-Free Survival, BF = Base Factors (sex, age, multiplicity, tumor size, stage, grade). BCG = Bacillus Calmette Guérin, EORTC = European Organization for Research and Treatment of Cancer. OS = Overall Survival, CSS = Cancer-Specific Overall Survival, DFS = Disease-Free Survival. NMIBC = Non-Muscle Invasive Bladder Cancer. AUC = Area Under the Curve, SE = Sensitivity, SP = Specificity, ↑ = Increased, ↓ = Decreased.

**Table 2 ijms-25-02178-t002:** Studies using miRNA-based models for BC stratification.

miRNAs Analyzed	Performance	Group Comparison	Type of miRNA	Source	Technique for Analysis	Sample Size	Validation	Authors and Year
let-7a-5p, -449a-5p, -124-3p, -138-5p and -23a-5p	↓ let-7a-5p, miR-449a-5p, -124-3p, and -138-5p = ↓ 1 and 5 yr. CSS and MIBC ↑ miR-23a-5p in MIBC vs. NMIBC	NMIBC vs. MIBC	Free	Tissue	RT-QPCR	100	No	Awadalla 2022 [37]
miR-21, -34a, -141, 193a, -200a and -200c	miR-34a, -193a and -200a classified high vs. low risk SE 0.88, SP 0.8, and ACC 0.82 ↑ All 6-miR expression = ↓ RFS	Low/intermediate risk vs. High risk (For recurrence)	Free	Urine and plasma	RT-QPCR	100	No	Cavallari 2020 [41]
**9 miRNA signature**	**Aggressive BCa =** **↓** **OS**	**Aggressive vs. non aggressive BC**	**Free**	**Tissue**	**Microarray** **TCGA analysis**	**87**	**Yes**	**Inamoto 2018 [43]**
**14 miRNA signature**	**Hypoxic =** **↓** **PFS and** **↓** **OS**	**Hypoxic MIBC vs. non-hypoxic MIBC**	**Free**	**Tissue**	**TCGA analysis**	**657**	**Yes**	**Khan 2021 [45]**
**7 miRNA-based score (-185-5p, -66a, -30c-5p, -3648, -1270, -200c-3p, and -29c-5p)**	**↑** **Score =** **↓** **OS**	**High score BC vs. Low score BC**	**Free**	**Serum**	**Microarrays**	**492**	**No**	**Lin 2019 [49]**
**Gene, lncmRNAs and miR-3913-1 and -981a score**	**↑** **Score =** **↓** **OS** **Score had** **↑** **AUC vs. TNM for survival**	**High score BC vs. Low score BC**	**Free**	**Tissue**	**TCGA analysis**	**239**	**No**	**Liu 2018 [52]**
**Genes, lncmRNAs and miR-497-5p**	**↑** **Score =** **↓** **OS**	**Low risk vs. low risk ** **(By score)**	**Free**	**Tissue**	**TCGA analysis**	**400**	**Yes**	**Liu 2020 [53]**
**7 miRNA-based score (-1247, -1304, -1911, -204, -33b, -3934, and -526b)**	**↑** **Score =** **↓** **OS** **AUC for 3–5-year survival 0.762**	**Low risk vs. High risk ** **(By score)**	**Free**	**Tissue**	**TCGA analysis**	**428**	**No**	**Liu 2020 [54]**
**miR-17-5p, 19a-3p and 19b-3p**	**↑** **Score =** **↓** **OS** **AUC 0.645 for progression**	**Low risk vs. High risk ** **(By score)**	**Free**	**Tissue**	**TCGA analysis**	**405**	**No**	**Pan 2020 [55]**
**Score lymph node + (miR-23a-3p, -3679-3p, ** **and -3195)**	**Score AUC .88, SE 0.87, SP 0.30 for recurrence** **↓** **Score =** **↑** **OS**	**High vs. Low index**	**Free**	**Tissue**	**RQ-QPCR**	**81**	**Yes**	**Urabe 2022 [60]**
**Clinical-mRNA-miRNA signature (miR-200c, -598 and -143)**	**CPV + signature =** **↑** **AUC and HR for** **↓** **5-year OS of both alone**	**Low risk vs. High risk ** **(By score)**	**Free**	**Tissue**	**TCGA analysis**	**402**	**No**	**Xiong 2018 [65]**
**7 miRNA signature ** **(-151-a-5p, -216a-5p, -337-3p, -let-7c, -125-b, -590-3p, 652-3p)**	**↑** **Score =** **↓** **OS and** **↑** **AUC of CPV**	**Low risk vs. High risk ** **(By score)**	**Free**	**Tissue**	**RT-QPCR** **TCGA Analysis**	**432**	**No**	**Xv 2022 [66]**
**21 miRNA signature**	**↑** **Score =** **↓** **OS**	**Low risk vs. High risk ** **(By score)**	**Free**	**Tissue**	**TCGA analysis**	**427**	**No**	**Yin 2019 [68]**

Note: Only statistically significant results are shown. Highlighted studies reflect those using multimarker models. BC = Bladder Cancer, RFS = Recurrence-Free Survival CPV = Clinicopathological variables (age, sex, grade, stage and TNM status). OS = Overall Survival, PFS = Progression-Free Survival, CSS = Cancer Specific Overall Survival. NMIBC = Non-Muscle Invasive Bladder Cancer, MIBC = Muscle Invasive Bladder Cancer. AUC = Area Under the Curve, SE = Sensitivity, SP = Specificity, ACC = Accuracy. ↑ = Increased, ↓ = Decreased.

**Table 3 ijms-25-02178-t003:** Studies identifying miRNAs as biomarkers for tumor progression in BC.

miRNAs Analyzed	Performance	Group Comparison	Type of miRNA	Source	Technique for Analysis	Sample size	Validation	Authors and Year
miR-9	↑ miR-9 in MIBC vs. NMIBC ↑ miR-9 in HG NMIBC vs. LG NMIBC	MIBC vs. NMIBC//LG NMIBC vs. HG NMIBC	Free	Tissue	RT-QPCR	90	No	Setti 2019 [57]
miR-222	↑ miR-222 in MIBC vs. NMIBC ↑ miR-222 in HG NMIBC vs. LG NMIBC ↑ miR-222 = ↓ RFS, ↓ DFS, ↓ PFS	Low expression vs. High expression	Free	Tissue	RT-QPCR	387	No	Tsikrika 2018 [59]
miR-133a, -143, and -200b	↓ miR-200b associated with MIBC	Low expression vs. High expression	Free	Tissue	Photonic crystal (PhC) barcodes with hybridization chain reaction (HCR)	10	No	Wei 2020 [63]
miR-10a-5p	↑ miR-10a-5p in MIBC vs. NMIBC AUC 0.78, SE 0.75, SP 0.64 for MIBC vs. NMIBC, ↓ OS and RFS	Low expression vs. High expression	Free	Tissue and plasma	RQ-QPCR	244	Yes	Yang 2021 [67]
miR-10a	↑ miR-10a = ↑ Grade and ↑ Stage	Low expression vs. High expression	Free	Tissue and urine	RT-QPCR	20	No	Zaidi 2023 [69]

Note: Only statistically significant results are shown. BC = Bladder Cancer, RFS = Recurrence-Free Survival. OS = Overall Survival, DSF = Disease-Free Survival. NMIBC = Non-Muscle Invasive Bladder Cancer, MIBC = Muscle Invasive Bladder Cancer. HG = High Grade, LG = Low Grade, AUC = Area Under the Curve, SE = Sensitivity, SP = Specificity. ↑ = Increased, ↓ = Decreased.

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
