# Peer review of "The Role of miRNAs to Detect Progression, Stratify, and Predict Relevant Clinical Outcomes in Bladder Cancer"

_ijms, 2024, doi:10.3390/ijms25042178_

Round 1

Reviewer 1 Report

Comments and Suggestions for Authors

In their manuscript, the authors review the current state of bladder cancer diagnostics and monitoring and highlight the possible future benefits of implementing miRNA signature detection into the clinical routine. They accurately point out the limitations of our current understanding of miRNAs’ (often two-faced) role in the disease and the lack of standardization in evaluation protocols.

My specific concerns about particular sections of the manuscript are detailed below.

In Chapter 1 (“Bladder cancer overview”), the detailed table on TNM staging seems like an overemphasis. Few of these stages have relevant relations to miRNA expression as explained in the later sections of the manuscript.

Line 93: This part is unclear and confusingly written. General understanding is that a near-perfect pairing results in mRNA degradation while moderate pairing results in translation inhibition without cleavage. This is most likely what the authors intended to write, although there is evidence of mRNA cleavage with only partial miRNA-target complementarity (see e.g. Xu K et al., MicroRNA-mediated target mRNA cleavage and 3-uridylation in human cells, 2016).

Again, I found the level of detail on general miRNA origins and function to be out of proportion. I recommend shortening it a bit.

Line 186: A difference in used methods is not the only factor that may account for contradictory findings on how a particular miRNA may affect clinical outcomes. Bladder cancer is a disease with considerable genetic heterogeneity, which probably affects miRNA profiles as well. Even if it is not possible to deal with such diversity within the constraints of the present review, I do feel that it should be mentioned at least as a factor limiting the interpretation of data (perhaps in Chapter 4, “Limitations of miRNA-based strategies for BCa”).

Knowledge (or lack thereof) on the two-faced roles of various miRNAs in bladder cancer is adequately highlighted in Chapter 3. I thought that if the authors provided some basic network analysis with figures on some known or proposed miRNA targets, they could also address the general lack of visual content, which is slightly unusual for a review.

References: reference 13 is a self-citation of an earlier paper by the same authors. The paper focuses on miRNAs’ role in prostate cancer, while the citation context is the formation of the RISC complex. Depending on journal policy, this may need to be removed.

Comments on the Quality of English Language

Use of the English language is generally adequate with minor errors and inconsistencies (e.g., “nuclei” should be in the singular in line 97).

Author Response

Dear Reviewer,

Thank you for your valuable comments. Here are the changes we made based on them:

-Table 1 has now been moved to supplementary material

-The confusion regarding near-perfect and moderate pairing has been addressed; please see lines 102-109

-We have shortened the discussion of miRNA biogenesis and functions

-We do agree that tumor heterogeneity is an important limiting factor; this has been included in section 4; please see lines 337-339

-We have created Figure 2 to illustrate the most consistent miRNAs in terms of genetic plausibility and the biological process they affect and implicate targets.

-Self-citation reference has been removed

Reviewer 2 Report

Comments and Suggestions for Authors

-       Bladder Cancer Overview: I suggest discussing in this first section the risk factors associated with bladder cancer, including the recent research on new biomarkers (doi: https://doi.org/10.1016/j.euros.2023.11.003).

-       Methodology and Result Validity: A more in-depth examination of the methodology employed in the 36 retrospective studies is essential. The validity of results may be influenced by the diversity in sampling methods, data analysis, and normalization techniques. Recommending a more thorough analysis can ensure more reliable outcomes.

-       Utilization of Liquid Biopsy: The use of liquid biopsies is intriguing, but the paper could delve further into this aspect, considering the advantages and challenges associated with miRNA analysis in urine or biological fluids compared to tissue samples.

-       Selection of Biomarkers: While the article mentions several miRNAs as promising biomarkers, a more thorough discussion on the rationale behind choosing specific miRNAs like the miR-200 and Let-7 families would be appreciated. A broader overview of biomarker selection would enhance understanding.

-       Limitations and Future Perspectives: Limitations of using miRNAs as biomarkers should be clearly discussed, along with potential future directions. This would contribute to a more balanced view of the potentials and challenges associated with the clinical application of miRNAs in bladder cancer.

Comments on the Quality of English Language

Moderate editing of English language required

Author Response

Dear Reviewer,

Thank you for your valuable comments. Here are the changes we made based on them:

-We have included the intriguing findings about the urobiome in the bladder cancer overview, the concept of liquid biopsy, and current biomarker research in BC; see lines 34-41 and 76-95

-The limitation section has been redesigned to discuss more in-depth methodology of the used studies; see lines 333-364

-The rationale underlying our selection of the miR-200 and Let-7 family members as candidates for new models is based on the consistent findings across studies, validation, and plausible mechanisms concordant with their expression levels in patient cohorts. This has been expanded in lines 380-389 and illustrated in Figure 2, which illustrates the pathways affected by these as well as Figure 3 of future directions.

-For future directions, we have reimagined Figure 3 to include future directions for new models or for studies reproducing already validated models, which has also been laid out in this section.

Reviewer 3 Report

Comments and Suggestions for Authors

This is a comprehensive review on the role of miRNAs in the prognosis and clinical outcomes of patients with bladder cancer. In my opinion, this review is comprehensive and informative, covering all aspects of the topic and providing a well-organized conclusion and future perspectives for the use of miRNAs in the prognosis of bladder cancer. However, I have a few concerns that need to be addressed:

1. The heading "miRNA as biomarkers in cancers" should be omitted, and the following parts should be added to the introduction of this review.

2. It is important to specify the role of miRNA in relation to the clinical staging or grading of bladder cancer, as well as the treatment response in patients with bladder cancer, as appropriate.

Comments on the Quality of English Language

Minor editing is required.

Author Response

Dear Reviewer,

Thank you for your valuable comments. Here are the changes we made based on them:

-We agreed that the heading of miRNAs as biomarkers in cancer is unnecessary; we have removed it and connected the two contents. Please see lines 102-109

-We included how miRNAs can also differentiate between tumor stage and grade. In addition, the tables were redesigned and reoriented, now separating studies used to stratify, predict clinical outcomes, or detect progression. Section 2 has been structured to include this in the discussion. Please see lines 153-156 and tables 1-3

Reviewer 4 Report

Comments and Suggestions for Authors

Thank you for letting me review this article. The authors aimed to review the role of miRNAs in Bladder cancer. My comments:

The abstract should be unstructured, Remove Background:, etc

The frequently used abbrev for bladder cancer is BC (also in nmiBC, miBC) and not BCa, please revise throughout the article.

I see you citing studies by Braicu which were of high quality, there was a publication in the IJMS reviewing miRNAs, I suggest reading it

https://www.mdpi.com/1422-0067/23/21/13206 - check your chosen miRs and if these correlate with your findings. Also, look at this article to complement the revision of sections 3-5 https://www.ncbi.nlm.nih.gov/pmc/articles/PMC5419229/.

Line 90 – miRNAs are 19-25, not 21-23, if you mean BC-specific miRNA then revise.

I suggest reworking tables 2 and 3 as they are blurry. – maybe use landscape page orientation for these specific sections

I suggest revising sections 3-5 for clarity of content.

Change the citation style adhering to journal guidelines.

Figure 1 should be earlier in text not after conclusions, this is not standard.

Comments on the Quality of English Language

typos and rewriting for clarity

Author Response

Dear Reviewer,

Thank you for your valuable comments. Here are the changes we made based on them:

-We have restructured the abstract and changed the in-text bladder cancer abbreviation.

-The papers of Homami and Harsanyi are certainly informative and confirm our findings of the miR200 having a tumor suppressor role in bladder cancer; this surely enriches our analysis in section 3

-We have corrected the miRNA "21-23" line, now present in line 102

-The tables will be changed to landscape orientation to make them clearer; these also were redesigned; see tables 1-3

-Sections 3 through 5 have been revised, and several corrections have been made

-The citation style has been addressed as per your and the editor's recommendations

-We have created some other figures: Figure 1, which summarizes our research process; Figure 2, which reveals the mechanisms of the most promising miRNAs; and Figure 3, which summarizes future directions

Reviewer 5 Report

Comments and Suggestions for Authors

I have reviewed the article titled 'The Role of miRNAs in Detecting Progression, Stratification, and Predicting Relevant Clinical Outcomes in Bladder Cancer' presented by Torres-Bustamente et al, regarding the involvement of miRNAs in various aspects of bladder cancer, and I have some observations that could help the authors improve the work.

The abbreviations in each table should be listed under the respective table, but this was missing in Table 1.

The authors mention that 'The only recommended biomarker in the guidelines is urine cytology, which can be used along cystoscopy' (lines 80-81). While this is the only recommended one, there are several reports suggesting the use of other biomarkers, which could be mentioned before discussing miRNAs. Please review doi: 10.3390/diagnostics10010039  https://doi.org/10.1002/ctd2.183

The paragraph from lines 80 to 85 needs to cite a reference.

Homogenize 'miRNA' or 'miRNA' throughout the text.

Tables 2 and 3 should begin with the column for the analyzed miRNAs followed by performance, and authors and year in the last column. The discussion regarding 'validated' and 'non-validated' miRNAs is not clear and tends to be redundant.

It wasn't entirely clear throughout the review which miRNAs were specifically implicated in detecting progression, stratification, and predicting relevant clinical outcomes of bladder cancer, as stated in the title. I suggest that these could be explained in subsections and tables for each aspect.

Figure 1 should have a closed caption and divide (part A and part B), explaining each one separately. Part A of the validated multimarker should add more specific examples of promising miRNAs, focusing only on let-7f-5p and the mir-200 family. Also, indicate the miRNAs involved in the clinicopathological variables.

The conclusions of the study should be better summarized.

Comments on the Quality of English Language

Moderate editing of English language

Author Response

Dear Reviewer,

Thank you for your valuable comments. Here are the changes we made based on them:

-Abbreviations have been included in all tables

-We have incorporated the FDA-approved biomarkers into the introduction section; please see lines 85-95

-The paragraph now has cited author in reference 16; please see lines 96-101

-miRNA has been homogenized in the whole manuscript

-Tables have been redesigned to be divided based on the purpose of each study, either predict clinical outcomes, stratify, or detect progression; please see tables 1-3

- Figure 1 has been reimagined; now, Figure 3 includes future directions for new models or studies reproducing already validated models, which has been laid out in the future directions section.

-The conclusions section was redesigned to be more concise 

Round 2

Reviewer 2 Report

Comments and Suggestions for Authors

I believe that the study has sufficient merit to be considered for publication

Author Response

Dear Reviewer:

Thank you for your feedback. Your valuable comments really enriched this project.

Reviewer 4 Report

Comments and Suggestions for Authors

My concerns have been addressed, the article is now more suitable for publication.

Author Response

(The authors gave the same response as above.)

Reviewer 5 Report

Comments and Suggestions for Authors

The authors have made significant changes to the revision, which, in the final decision of the Editor, is considered suitable for publication.

Author Response

(The authors gave the same response as above.)
